# Investigation of the Efficacy of Pyrantel Pamoate, Mebendazole, Albendazole, and Ivermectin against *Baylisascaris schroederi* in Captive Giant Pandas

**DOI:** 10.3390/ani13010142

**Published:** 2022-12-29

**Authors:** Yaxian Lu, Linhua Deng, Zhiwei Peng, Mengchao Zhou, Chengdong Wang, Lei Han, Shan Huang, Ming Wei, Rongping Wei, Lihong Tian, Desheng Li, Zhijun Hou

**Affiliations:** 1College of Wildlife and Protected Area, Northeast Forestry University, Harbin 150040, China; 2China Conservation and Research Centre for the Giant Panda, Dujiangyan 611843, China

**Keywords:** *Ailuropoda melanoleuca*, anthelmintic administration, anthelmintic resistance, FECRT

## Abstract

**Simple Summary:**

Giant pandas are key protected animals in China. *Baylisascaris schroederi*, a parasitic nematode, is one of the main health risks threatening them. We used four anthelmintics—pyrantel pamoate (PYR), mebendazole (MBZ), albendazole (ABZ), and ivermectin (IVM)—on 22 enrolled giant pandas. The fecal egg count reduction (FECR) proportions were calculated using both the Markov chain Monte Carlo (MCMC) Bayesian mathematical model and the arithmetic mean based on fecal egg count data. Anthelmintic resistance (AR) was assessed based on the criteria recommended by the World Association for the Advancement of Veterinary Parasitology (WAAVP). We found that the nematode was suspected to be resistant to PYR. The number of eggs per gram in the feces of giant panda enrolled in the present study was increased near the end of the experiment.

**Abstract:**

*Baylisascaris schroederi* is one of the main health risks threatening both wild and captive giant pandas. The administration of anthelmintics is a common method to effectively control *B. schroederi* infection, but there is a notable risk of anthelmintic resistance (AR) after long-term, constant use of anthelmintics. Four anthelmintics—pyrantel pamoate (PYR), mebendazole (MBZ), albendazole (ABZ), and ivermectin (IVM)—were each administered separately at intervals of 2 months to 22 enrolled giant pandas. The fecal egg count reduction (FECR) proportions were calculated by both the Markov chain Monte Carlo (MCMC) Bayesian mathematical model and the arithmetic mean. AR was assessed based on the criteria recommended by the World Association for the Advancement of Veterinary Parasitology (WAAVP). The estimated prevalence of *B. schroederi* infection was 34.1%. After treatment with PYR, MBZ, ABZ, and IVM, it was determined that MBZ, ABZ, and IVM were efficacious against *B. schroederi*, while nematodes were suspected to be resistant to PYR according to the fecal egg count reduction (FECR) proportions.

## 1. Introduction

The giant panda (*Ailuropoda melanoleuca*), a flagship species for conservation, is restricted to natural habitats in western China [1]. Habitat loss, degradation and fragmentation, poor reproduction, and limited resistance to some infectious diseases threaten this species [2,3,4]. To protect giant pandas, more than 375 individuals have been raised at conservation centers or in zoos [5]. Of these factors, diseases caused by parasites, especially *Baylisascaris schroederi*, are reported to be one of the main health risks threatening giant pandas in the wild and at conservation centers [4,6].

*Baylisascaris schroederi*, first reported by Mcintosh (1939) [7], can cause lethargy, inappetence, malnutrition, anemia, pancreatitis, and even death as a result of intestinal impaction and subsequent rupture in giant pandas with a high *B. schroederi* parasite load [6,8].

Current methods available for the control of ascarid infection rely predominantly on the periodic administration of anthelmintic drugs [9,10]. As the prevalence of *B. schroederi* is very high among captured giant pandas in zoos in China, these pandas undergo deworming treatment at regular intervals using at least one registered anthelmintic drug in two to six rotational treatments a year. This long-term, high-frequency anthelmintic drug use leads to considerable selective pressure on parasites and creates a high risk of emerging drug resistance [11].

However, research on anthelmintic resistance (AR) is very limited in giant pandas; there have been only two reports on this topic, both of which used the cure proportion method to identify resistance [12,13]. Pyrantel pamoate (PYR) and ivermectin (IVM) failed to achieve full efficacy, while mebendazole (MBZ) and albendazole (ABZ) were fully efficacious. However, the AR level could not be interpreted from the available data in the two studies.

Given the fear that AR emergence caused by selective pressure from drugs will eventually hinder deworming interventions against *B. schroederi*, monitoring of *B. schroederi* infections to track AR is necessary for optimizing treatments and reducing further selection.

The McMaster technique, quantifying the number of eggs in a fecal sample, is a standard tool for diagnostic parasitologists and has been widely used in veterinary medicine for assessing AR in parasites of domestic animals [14,15].

To achieve an understanding of drug efficacy and determine the presence or absence of AR, the present study aimed to estimate the prevalence of *B. schroederi* infection and assess the efficacy of PYR, MBZ, ABZ, and IVM, the helminthics registered to combat *B. schroederi* infection in giant pandas.

## 2. Materials and Methods

The study was carried out in the Dujiangyan Base of the China Conservation and Research Center for the Giant Panda (CCRCGP), one of the largest giant panda centers in China, which has a history of rescuing giant pandas for nearly 10 years.

All giant pandas that were sick, pregnant, or lactating were excluded; the remaining 22 of them (male = 10, female = 12) were included in the present study and housed in individual yards. According to the principles of age classification of giant pandas, the 22 pandas were divided into three age groups: juvenile (1.5–5.5 years old, N = 10), adult (5.5–10 years old, N = 3), and geriatric (20–27 years old, N = 9). The giant pandas underwent natural infection with *B. schroederi*. From April to October 2018, fecal samples were collected four times (every 2 months) each time after PYR, MBZ, ABZ, and IVM were separately administered in turn to the 22 enrolled giant pandas at intervals of 2 months (Table 1).

In each deworming treatment, paired fecal samples (pretreatment and post-treatment) were collected on 11 different days (Table 2); placed in marked, airtight plastic bags, and immediately presented to the laboratory. Adult worms were collected within 3 days after the giant pandas were dewormed by their keepers.

Subsequently, the fecal egg count (FEC) was performed with a modified McMaster technique (the minimum egg detection limit was 24 eggs per gram (EPG), measured with 100 g of feces and 260 mL of saturated sodium chloride solution) for the quantification of *B. schroederi* eggs [16]. The samples that tested negative by the modified McMaster technique were retested using the saturated saline flotation method to determine whether they were truly negative or whether they simply had false-negative results because their egg counts were below the minimum egg detection limit of the McMaster technique. From each sample, duplicate FEC data were obtained, and the average was used to calculate the FECR to improve the precision of egg counts. The prevalence of patent *B. schroederi* infections was calculated as the number of animals testing positive divided by the total number of giant pandas sampled.

The fecal egg count reduction test (FECRT) was performed to calculate the reduction proportion. The fecal egg count reduction% (FECR%) = 100(1 − X_post_/X_pre_) (X_pre_: EPG before anthelmintic treatment on Day 0; X_post_: EPG post anthelmintic treatment on Day 15) and its 95% confidence interval (95% CI) were calculated at the group level using two approaches: the arithmetic mean and a Markov chain Monte Carlo (MCMC) Bayesian approach. The approach was based on the WAAVP (World Association for the Advancement of Veterinary Parasitology) guidelines on AR but also considered the upper 95% confidence limit (95% CL_U_), as well as the lower 95% confidence limit (95% CL_L_), and the percentage reduction was taken to interpret the AR of *B. schroederi* [15,17,18]. Anthelmintic efficacy was classified into three levels based on the following criteria [10,15,17,18,19,20,21]:(i)Efficacy, where FECRs and the 95% CL_U_ were >95% and the 95% CL_L_ was >90%;(ii)Resistance, where FECRs and the 95% CL_U_ were <95% and the 95% CL_L_ was <90%;(iii)Suspected resistance, where none of the above conditions were met.

The arithmetic mean model implemented in fecrtCI (eggCounts 2.1–2 package) in R (version 3.6.0) and the MCMC Bayesian hierarchical model implemented in the fecr_stan tool (eggCounts 2.3 package) in R (version 4.0.3) were adopted [22,23].

All statistical studies were conducted using univariate analysis (the Wilcoxon test) and the “ggpubr” package in R (version 4.0.2) [24].

## 3. Results

A total of 88 fecal samples from 22 giant pandas (four replicates per animal) were collected before treatment on Day 0, and ascarid eggs were detected in 30 of the fecal samples, resulting in an estimated 34.1% prevalence of *B. schroederi* infection (Table 2).

As calculated by the arithmetic mean, the FECR was 94.9% (95% CI: 51.2–99.5%) when the giant pandas were treated with PYR, and it was 100% with MBZ, ABZ, and IVM. Meanwhile, as calculated by the MCMC Bayesian approach, the FECRs were 87.4% (95% CI: 54.5–99.6%), 99.8% (95% CI: 99.3–100%), 99.80% (95% CI: 98.8–100%), and 99.6% (95% CI: 98.3–100) after treatment with PYR, MBZ, ABZ, and IVM, respectively. In the present study, MBZ, ABZ, and IVM were determined to be efficacious, and the nematode was suspected to be resistant to PYR. The quantity of EPG was increased near the end of the experiment (Table 2).

A total of 255 adult worms were collected, 60.4% of which were collected after the giant pandas were administered PYR (Figure 1). In addition, most of the worms were from male pandas, and the numbers of worms from the male and female hosts were significantly different (Figure 2). There was an interesting phenomenon in which the sex ratio of the *B. schroederi* collected after deworming in the present study was approximately 3:1 (female:male) (Figure 3).

## 4. Discussion

This is the first report on the possibility of PYR resistance in *B. schroederi* and provides preliminary insight into the current state of AR in captive giant pandas in China. There are some reports of PYR resistance in *Parascaris* spp. [20,25,26,27]. By the interpretation criteria used in those reports, *B. schroederi* was also suspected to be resistant to PYR in the present study. Although *B. schroederi* was resistant to PYR by some other rules (Table 3), it is more plausible that *B. schroederi* is merely suspected to be resistant, as only 1/7 of animals treated with PYR failed to become negative 15 days after treatment. Overall, these results suggested with an extremely high likelihood that some degree of AR to PYR existed in *B. schroederi*, and additional regular surveillance and a sensitive molecular detection method are necessary to monitor the status of PYR resistance in *B. schroederi* in the future.

Current evidence from equine and ruminant nematodes suggests that fitness is not significantly compromised in drug-resistant strains. The drug-resistant strains would not be likely to revert to susceptibility even if they were left unexposed to the anthelmintic for a long period [11,28,29]. Thus, it is reasonable to presume that AR has little likelihood of disappearing spontaneously; that is, reversion to anthelmintic sensitivity is unlikely to occur once *B. schroederi* has become resistant. In the absence of alternative treatments, anthelmintic drugs remain the major method of controlling parasitic nematode infections in the short to medium term, but AR threatens the sustained efficacy of the limited number of available drugs. However, new anthelmintics are not expected to solve this problem, owing to the long time investment and excessive cost of introducing a completely new drug on the market [30]. Regarding the giant panda, although it is a species that attracts abundant attention from researchers and conservationists, the exploration of new drugs for this species has far fewer prospects, as there is almost no commercial interest in the giant panda within the pharmaceutical industry. Consequently, it is imperative that anthelmintics registered for giant panda use be administered judiciously and that the further development of AR in *B. schroederi* be avoided or delayed as much as possible.

IVM was determined to be effective in the present study, but it is an anthelmintic to which ascarids readily build resistance under a certain degree of selective pressure [31]. One of the possible reasons for this is that IVM has larvicidal efficacy against ascarids in the liver and lungs and could lead to the removal of a source of refugia otherwise exploited by migrating larvae, in contrast to anthelmintics with no activity against larvae [31,32]. Therefore, it will generate stronger selective pressure toward AR than other drugs. A possible sign of AR in *B. schroederi* appeared in one of our studies: the AR-related glc-1- and pgp-3 genes were inferred to be undergoing selection from IVM administration in *B. schroederi* [33]. Monitoring the efficacy of IVM needs to be a high priority for practitioners in giant panda conservation.

The available approach to detecting the AR of ascarid infections in veterinary medicine is the FECRT. However, it should be noted that the FECRT was generally developed for detecting AR in ruminant nematodes, and a universal threshold for defining AR in ascarids, even in the ones studied in greater depth, such as *Parascaris* spp. and Ascaris lumbricoides, has yet to be agreed upon [11,21]. As the feces of the giant panda contain an abundance of undigested bamboo, this material will reduce the relative quantity of EPG and lower the sensitivity of the McMaster technique in pandas compared to ruminants or horses. Only when conducted in the field is the FECRT a reliable and suitable test for assessing AR, and no preliminary work has been performed to establish the optimum threshold for defining AR in *B. schroederi*.

Furthermore, the FECRT is reliable only when more than 25% of the nematode worms in a given population are resistant [14,34]. Thus, its utility in assessing drug efficacy is uncertain, and detecting the early stage of AR emergence in *B. schroederi* is not possible with the FECRT. Overall, an accurate diagnostic tool capable of detecting resistance, especially at an early stage, is urgently needed, as it will help conservationists notice the appearance of AR in *B. schroederi* and adopt suitable measures to slow the development of AR before therapeutic failure. A highly sensitive molecular technique capable of detecting the genotypic resistance of *B. schroederi* could resolve this issue.

The extensively used anthelmintic ABZ has been shown to have different degrees of pharmacological efficacy in hosts of different sexes, and most anthelmintics produce larger effects in male hosts than in female hosts [35]. In the present study, significantly more worms were collected from male giant pandas than from females after the administration of PYR, but there was no significant sex difference in the effect of ABZ, IVM or MBZ. It is believed that biological sex affects physiological and immune responses and thus drug metabolism and disease progression [36,37]. However, it is not necessarily true that all anthelmintics have different pharmacodynamics between the sexes [38]. There is no reasonable explanation for the phenomenon at present.

Although some measures, such as maintaining good hygiene and using anthelmintic treatments rotationally at bimonthly intervals, have been implemented, giant pandas still inevitably experience natural infection with *B. schroederi* as they are continuously exposed to environments contaminated with ascarid eggs. The problem of reinfection in panda colonies is still the greatest issue. Therefore, along with anthelmintic administration, stricter measures to maintain good hygiene are needed, as are regulations regarding inspection, quarantine, and isolation protocols for new animal residents to avoid introducing new sources of *B. schroederi*.

## 5. Conclusions

MBZ, ABZ and IVM were determined to be efficacious against *B. schroederi*, while nematodes were suspected to be resistant to PYR according to the fecal egg count reduction (FECR) rates. The number of eggs per gram was increased near the end of the experiment.

## Figures and Tables

**Figure 1 animals-13-00142-f001:**
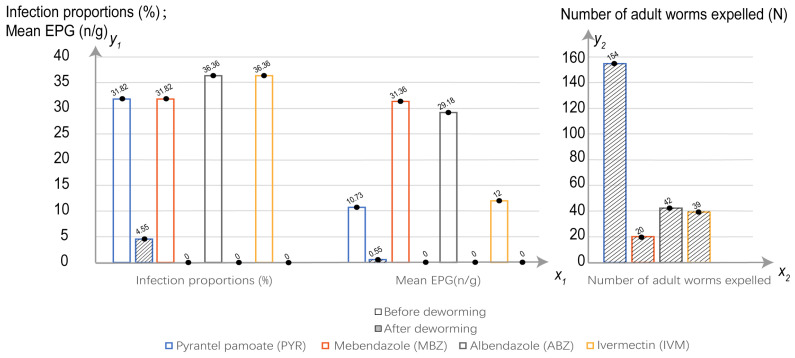
Infection rate, mean EPG, and number of adult worms.

**Figure 2 animals-13-00142-f002:**
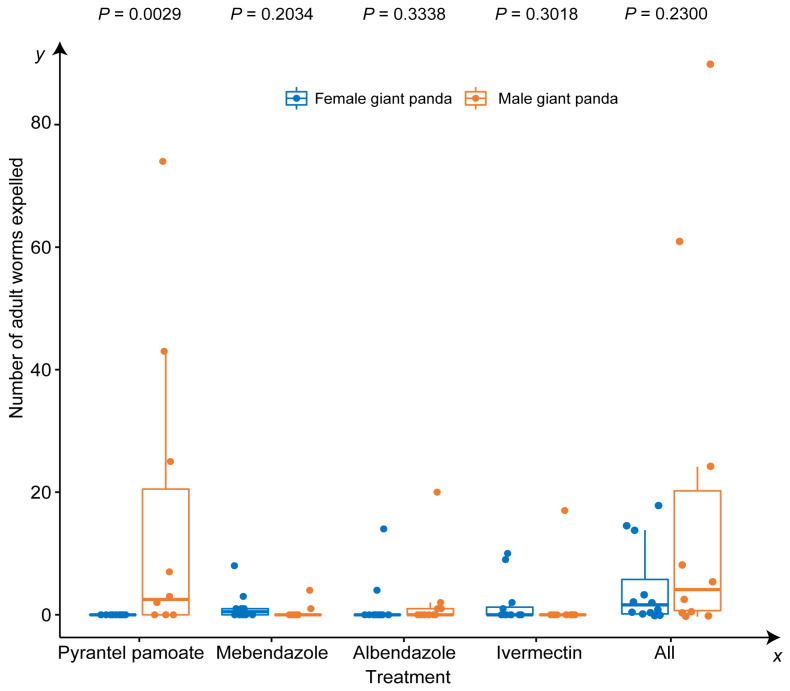
Mean number of worms expelled by different sexes of giant pandas after administration of four anthelmintics.

**Figure 3 animals-13-00142-f003:**
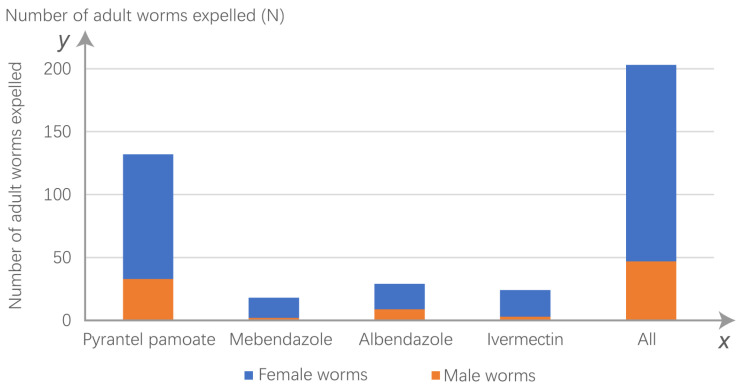
Sex ratio of expelled worms.

**Table 1 animals-13-00142-t001:** Anthelmintic drugs used for treatment of *B. schroederi* in giant pandas.

Anthelmintic Drug	Monitoring Interval (2018)	Dosage	Treating Day
PYR (Bimeda Inc., paste, Dublin, Ireland)	25 April–24 June	0.1 g/kg BW *	24 April
MBZ (Xian-Janssen Inc., paste, Xi’an, China)	25 June–21 August	10 mg/kg BW *	24–26 June
ABZ (SK&F Inc., paste, Tianjin, China)	22 August–23 October	10 mg/kg BW *	21–23 August
IVM (Vetone Inc., paste, Boise, America)	24 October–20 December	0.3 mg/kg BW *	23–24 October

* BW: body weight.

**Table 2 animals-13-00142-t002:** The infective intensities and proportions of *Baylisascaris schroederi* before and after treatment with pyrantel pamoate (PYR), mebendazole (MBZ), albendazole (ABZ), or ivermectin (IVM) for the captive giant panda.

Drug		Day 0	Day 1	Day 4	Day 8	Day 15	Day 22	Day 29	Day 36	Day 43	Day 50	Day 57
PYR	mean EPG (n/g)	10.7	2.2	0.1 ^a^	2.2	0.6	0.8	0.3	0.6	3.6	7.4	31.4
	infection rate (%)	31.8% (7/22)	27.3% (6/22)	4.5% (1/22)	4.6% (1/22)	4.6% (1/22)	4.6% (1/22)	4.6% (1/22)	9.1% (2/22)	18.2% (4/22)	36.4% (8/22)	31.8% (7/22)
MBZ	mean EPG(n/g)	31.4	6.8	0.1 ^a^	0	0	0	0	0	0.3 ^b^	3.0	29.2
	infection rate (%)	31.8% (7/22)	40.9% (9/22)	4.5% (1/22)	0%	0%	0%	0%	0%	13.6% (3/22)	40.9% (9/22)	36.4% (8/22)
ABZ	mean EPG(n/g)	29.2	\	\	\	0	\	\	\	\	1.1	12
	infection rate (%)	36.4% (8/22)	\	\	\	0%	\	\	\	\	13.6% (3/22)	36.4% (8/22)
IVM	mean EPG(n/g)	12	0.01 ^a^	0	0	0	0	0	0	0	0.1 ^a^	0
	infection rate (%)	36.4% (8/22)	4.6% (1/22)	0%	0%	0%	0%	0%	0%	0%	4.6% (1/22)	0%

^a^ One of the feces samples was found to be positive by detecting using the method of saturated saline flotation after it was detected as negative by the method of McMaster. ^b^ Three of the feces samples were found to be positive by detecting with the method of saturated saline flotation after they were detected as negative by the method of McMaster.

**Table 3 animals-13-00142-t003:** The anthelmintic efficacy as assessed by the arithmetic mean and MAMC Bayesian approach with different rules.

Rule *	Anthelmintic Resistance (AR)	Efficacy	Suspected Resistance (Inconclusive)	Interpretation Results Based on the FECRT Data of the Present Research				
				Approach	PYR	MBZ	ABZ	IVM
Rule 1 [17,18]	FECR < 95% and CL_L_ < 90%, and CL_U_ < 95%	FECR > 95% and CL_L_ > 90%, and CL_U_ > 95%	Neither of other two criteria	Arithmetic Mean	Suspected Resistance	Efficacious	Efficacious	Efficacious
				MCMC Bayesian	Suspected Resistance	Efficacious	Efficacious	Efficacious
Rule 2 [15]	FECR < 95% and CL_L_ < 90%	FECR > 95% and CL_L_ > 90%	Neither of other two criteria	Arithmetic Mean	Resistance	Efficacious	Efficacious	Efficacious
				MCMC Bayesian	AR	Efficacious	Efficacious	Efficacious
Rule 3 [19]	FECR < 90% and CL_L_ < 90%	FECR ≥ 95% and CL_L_ > 90%	FECR ≤ 90% or CL_L_ < 90%	Arithmetic Mean	Suspected Resistance	Efficacious	Efficacious	Efficacious
				MCMC Bayesian	AR	Efficacious	Efficacious	Efficacious
Rule 4 [20]	FECR < 80% and CL_L_ < 90%	FECR > 95% and CL_L_ > 90%	FECR in 80–90% and CL_L_ < 90%	Arithmetic Mean	Null	Efficacious	Efficacious	Efficacious
				MCMC Bayesian	Suspected Resistance	Efficacious	Efficacious	Efficacious
Rule 5 [10,21]	FECR < 95% and CL_U_ < 95%	CL_L_ > 95%	FECR including 95% in their 95% CI	Arithmetic Mean	Suspected Resistance	Efficacious	Efficacious	Efficacious
				MCMC Bayesian	Suspected Resistance	Efficacious	Efficacious	Efficacious

CI: Confidence interval. CL_U_: upper 95% confidence limits. CL_L_: Lower 95% confidence limits. * Rule for determining anthelmintic resistance.

## Data Availability

Not applicable.

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
