# Peer review of "Investigation of the Efficacy of Pyrantel Pamoate, Mebendazole, Albendazole, and Ivermectin against Baylisascaris schroederi in Captive Giant Pandas"

_animals, 2022, doi:10.3390/ani13010142_

Round 1

Reviewer 1 Report

The authors of the manuscript presented a study on the effectiveness of 4 different antiparasitic drugs against Baylisascaris schroederi in captive giant pandas.  The importance of this work is that such a study was conducted for the first time in giant pandas.

I have the following comments and/or suggestions:

1. I understand that giant pandas are a rare and valuable animal species, however, 22 animals is a small group. I think this is an important limitation of this study. Please address this issue.

2. Lines 100-110: Please describe in a little more detail the McMaster and the flotation methods that were used.

3. Lines 103-106: Do I understand correctly that the negative sample in the McMaster method was tested using the flotation method for confirmation? According to my knowledge, the McMaster Method is more sensitive than the flotation method. Please clarify/correct this point.

4. Frequently, pyrantelum is less effective in the treatment of roundworms compared to mebendazole, albendazole or ivermectin. Therefore, please justify in the manuscript the orgnality of your investigations.

5. In conclusion, please also include your findings on the effectiveness of antiparasitic drugs.

Author Response

  1. I understand that giant pandas are a rare and valuable animal species, however, 22 animals are a small group. I think this is an important limitation of this study. Please address this issue.

Answer1: As the reviewer stated, the giant panda is a rare animal species. The number of the giant pandas can be enrolled is limited, it maybe affects the AR assessing result. But to assessing the AR status of B. schroederi is needed, so we selected the FECT with a pair test to overcome the flaw as much as possible when no other method is available. The AR assessing of the drugs in the present study is that B. schroederi was suspected to be resistant to PYR, we call on a highly sensitive molecular technique capable of detecting genotypic resistance of to resolve this issue in the future in the discussion.

  1. Lines 100-110: Please describe in a little more detail the McMaster and the flotation methods that were used.

Answer2: The technique of McMaster is that taking 100 grams` feces of giant panda in a beaker, added 260 ml saturated sodium chloride solution and mixed them. Then, filtering the debris away with a sieve of 40 mesh and taken the filtered fluid into the chamber of the McMaster to count the number of eggs.

The flotation method is that taking about 100 grams` feces of giant panda in a beaker, added about 100 ml saturated sodium chloride solution and mixed them. Then, filtering the debris away with a sieve of 40 mesh and the filtered fluid were transferred into another tuber. Keeping it standing for 20 minutes and taken the upper surface fluid with the 1000 ul pipette tip (root open) onto the glass slide and observed in the microscope.

  1. Lines 103-106: Do I understand correctly that the negative sample in the McMaster method was tested using the flotation method for confirmation? According to my knowledge, the McMaster Method is more sensitive than the flotation method. Please clarify/correct this point.

Answer3: Yes, the negative sample in the McMaster method was tested using the flotation method for confirmation in the present study. In the present study, the McMaster method taken 100 grams’ feces and 260 ml saturated sodium chloride solution, while the flotation method only taken about 100 grams’ feces and 100 ml saturated sodium chloride solution, therefore the flotation method is more sensitive than McMaster method.

  1. Frequently, pyrantelum is less effective in the treatment of roundworms compared to mebendazole, albendazole or ivermectin. Therefore, please justify in the manuscript the orgnality of your investigations.

Answer4: Do we understand correctly that the orgnality is originality? Although PYR is usually less effective in the treatment of roundworms, however, in the practice of deworming for the giant panda, PYR could kill the B. schroederi more effectively that other drugs sometimes. Such as in the present study, the adult worm collected after treatment is most much with PYR (Pig. 2 and 3).

  1. In conclusion, please also include your findings on the effectiveness of antiparasitic drugs.

Answer5: We added the related content in the conclusion in the revised MS.

Reviewer 2 Report

Dear authors, 

I reviewed the manuscript animals-2090792 entitled "Investigation of the efficacy of pyrantel pamoate, mebendazole, albendazole, and ivermectin against Baylisascaris schroederi in captive giant pandas" by Yaxian et al. 

The manuscript reports an investigation about the efficacy of four antiparasitic drugs against Baylisascaris schroederi in giant pandas.

In my opinion the subject of the study falls within journal’s scope and results have interest to readers, however there is some shortcomings that should be addressed.

Therefore, I suggest to consider the manuscript for publication pending minor revisions. 

Specific comments are listed below: 

I recommend to include giant panda scientific name in the title or in the keywords of the paper.

L30-32: Please rewrite the sentence “were determined to be efficacious against B. schroederi, while…”

L39-40: I suggest to rewrite: “is restricted to western China natural habitats”

L39-43: Include in this paragraph which percentage of giant pandas life in a natural habitat/captivity.

L47: Baylisascaris schroederi

Table 1: Remove “four” → “Anthelmintic drugs used for treatment of B. schroederi in giant pandas”

Table 1: *BW: body weight.

In my opinion Table 2 should not be in that location.

Table 2: what is the meaning of the symbol “\”?

L128: include the reference of the “ggpubr” package.

Figure 1:

Remove “the” from the figure title.

Include the axis X and Y (both) titles.

Which values are associated to each Y axis (right and left)?

Include the complete name of the drugs.

The differences between before and after treatment columns are difficult to understand. Please use other colors (i.e. white pattern instead of the black one).

Figure 2 and 3:

Write the axis title names with initial capital letters.

Title Y axis. Please rewrite “Number of adult worms expelled”

Remove “the sex of giant pandas” and “the sex of worms” and rewrite for: “female giant panda” and “male giant panda” and “male worms” and “female worms”.

I also suggest to use the same color for female pandas and nematodes (i.e. blue for female pandas and nematodes and orange for male pandas and nematodes).

Please correct the references’ style according to authors’ guidelines.

Author Response

  1. The comments of

1) I recommend to include giant panda scientific name in the title or in the keywords of the paper, 2) L30-32: Please rewrite the sentence “were determined to be efficacious against B. schroederi, while…”, 3) L39-40: I suggest to rewrite: “is restricted to western China natural habitats”, 4) L47: Baylisascaris schroederi

Answer1: All of them were accepted in the revised MS.

  1. L39-43: Include in this paragraph which percentage of giant pandas life in a natural habitat/captivity.

Answer2: The pandas in the present study refers the individuals who were captured in

 the zoo or conservation centers, none of them is in natural habitat/captivity. There

 were 1864 individuals in the natural habitat in 2015 (Tang XP, 2015).

Table

  1. 1) Remove “four” → “Anthelmintic drugs used for treatment of B. schroederi in giant pandas”, 2)Table 1: *BW: body weight

Answer3: Both of them were accepted in the revised MS.

  1. In my opinion Table 2 should not be in that location.

Answer4: The table 2 has been moved in a new location in the revised MS.

  1. Table 2: what is the meaning of the symbol “\”?

Answer5: It means the data missed.

  1. L128: include the reference of the “ggpubr” package.

Answer6: The reference 24 was added in the revised MS.

Figure 1:

  1. Those comments of 1) Remove “the” from the figure title, 2) Include the axis X and Y (both) titles, 3) Include the complete name of the drugs, 4) Please use other colors (i.e. white pattern instead of the black one)

Answer7: All comments were accepted in the revised MS.

  1. Which values are associated to each Y axis (right and left)?

Answer8: The picture 1 is a recombination picture, the Y1 indicated the values

of infection intensity (n/g) and infection proportion (%); the Y2 indicated the values of number of adult worms expelled from the giant panda.

  1. Include the complete name of the drugs.

Answer9: The complete name of the drugs was added in the revised MS.

  1. The differences between before and after treatment columns are difficult to understand. Please use other colors (i.e. white pattern instead of the black one).

Answer10: the columns are vaguely and they are instead by clear ones in revised MS.

Figure 2 and 3:

  1. The comments of 1) Write the axis title names with initial capital letters, 2)

Title Y axis. Please rewrite “Number of adult worms expelled”, 3) Remove “the sex of giant pandas” and “the sex of worms” and rewrite for: “female giant panda” and “male giant panda” and “male worms” and “female worms”, 4) I also suggest to use the same color for female pandas and nematodes (i.e. blue for female pandas and nematodes and orange for male pandas and nematodes)

Answer11: All of them were accepted in the revised MS.

  1. Please correct the references’ style according to authors’ guidelines.

Answer12: We check the references style according the journal require.

Reviewer 3 Report

Simple Summary and or Abstract – please mention that Baylisascaris schroederi is a parasitic nematode

Why this arrangement “pyrantel pamoate (PYR), mebendazole (MBZ), albendazole (ABZ), and ivermectin (IVM)” and not “albendazole (ABZ), ivermectin (IVM), mebendazole (MBZ), and pyrantel pamoate (PYR)”, i.e. alphabetic order?

Line 15 – FECR are not rates but proportions -change accordingly throughout the manuscript

Simple Summary – define AR

English needs a general review, preferably by a native speaker of the language – this reviewer will not point out all these aspects

Lines 18-20 – not clear, please rewrite

Line 25 – 2 months

Line 30 – use one decimal place only

Line 30 – insert comma between IVM and MBZ

Line 31 – while nematodes (instead of while the nematodes)

Lines 32-35 – these conclusions are not based on the results – please adapt

Lines 41-42 – replace semicolons with commas

Introduction – reduce the number of paragraphs by merging them

Line 44 – better define B. schroederi at its first mention

Line 51 – delete “mass”

Line 58 – not a true rate but a proportion

Materials and Methods – question: why did the authors not use controls?

Line 78 – 10 years

Line 79 – how many pandas were excluded?

Line 84 – four times

Line 85, etc. – 2 months

Line 89 – Adult worms were collected…

Table 2. Eggs per gram (EPG) and proportion (%) of positive animals…

Use only one decimal place

Table is incomplete for Albendazole – it would be better to exclude this drug

Liner 112 – replace rate with value or proportion

Line 113 – not only on Day 15

Line 115 – which one of the two approaches has been presented in Table 2?

Line 119 – this is not true efficacy (which would have involved a control)

Results – use only one decimal place

Figure 1 – delete rate; what is the meaning of “g”? Before treatment column is confusing

Table 3. The anthelmintic efficacy assessed by arithmetic mean…

Criteria (instead of criterions); Interpretation…

Line 171 – Current evidence from…

Conclusions - they should be more focused on the results

References – adapt to the style of Animals – not sure whether titles need to be between quotation marks, use lowercase as much as possible, etc.

Ref. 5 – is there an abstract in English?

Author Response

1.Simple Summary and or Abstract – please mention that Baylisascaris schroederi is a parasitic nematode.

Answer1: We have rewritten the sentence as “Baylisascaris schroederi, a parasitic nematode, is one of the main health risks threatening them.” in the revised MS.

  1. Why this arrangement “pyrantel pamoate (PYR), mebendazole (MBZ), albendazole (ABZ), and ivermectin (IVM)” and not “albendazole (ABZ), ivermectin (IVM), mebendazole (MBZ), and pyrantel pamoate (PYR)”, i.e. alphabetic order?

Answer2: The order of it is sequence of anthelmintic given to giant panda.

  1. Line 15 – FECR are not rates but proportions -change accordingly throughout the manuscript.

Answer3: We have instead of rates by proportions in whole MS

  1. define AR.

Answer4: We rewrite the sentence in L17-18 as “Anthelmintic resistance (AR) was assessed based on…”

  1. English needs a general review, preferably by a native speaker of the language – this reviewer will not point out all these aspects

Answer5: The manuscript has been polished by an English Language Service through tout the manuscript.

  1. Lines 18-20 – not clear, please rewrite

Answer6: We have rewritten as “We found that nematode was suspected to be resistant to PYR according to the fecal egg count reduction (FECR) proportions. The numbers of eggs per gram in the feces of giant panda enrolled in the study were increased near the end of the experiment.” In L19-21.

  1. Line 25 – 2 months; Line 30 – use one decimal place only; Line 30 – insert comma between IVM and MBZ; Line 31 – while nematodes (instead of while the nematodes); Lines 41-42 – replace semicolons with commas; better define B. schroederi at its first mention; Line 51 – delete “mass”; Line 58 – not a true rate but a proportion.

Answer7: All of them were accepted in the revised MS.

  1. Lines 32-35 – these conclusions are not based on the results – please adapt

Answer8: We have revise related content. The paragraph of “Although some measures, such as maintaining good hygiene ….. new animal residents to avoid introducing new sources of B. schroederi.” , has been moved to the section of discussion in the last paragraph in L223-230.

A new conclusion of “MBZ, ABZ and IVM were determined to be efficacious against B. schroederi, while nematode was suspected to be resistant to PYR. The number of eggs per gram in the feces of giant pandas enrolled in the present study were increased near the end of the experiment”was concluded in L232-235.

  1. Introduction – reduce the number of paragraphs by merging them

Answer9: Eight paragraphs have been merged to five paragraphs in the section of introduction of the revised MS.

  1. 1) Materials and Methods – question: why did the authors not use controls? ,

2) Line 119 – this is not true efficacy (which would have involved a control)

Answer10: There are three field designs namely (I) post-drench FEC of treatment and controls groups, (II) pre- and post-drench FEC of a treatment group only and (III) pre- and post-drench FEC of treatment and control groups (Lyndal-Murphy, Swain et al. 2014). Accessing the AR for the giant panda is difficult and no completely rational method is available. Although the methods of I and III with control are better than one of II, as number of the giant pandas can be enrolled is limited, we selected the method of II in the present study. Too small giant panda number maybe influence the AR assessing result, therefore we just concluded that “ nematode was suspected to be resistant to PYR” and call on a highly sensitive molecular technique capable of detecting genotypic resistance of B. schroederi to resolve this issue in the future in the discussion.

  1. Those comments of 1) Line 78 – 10 years, 2) Line 84 – four times, 3) Line 85, etc. – 2 months, 4) Line 89 – Adult worms were collected, 5) Table 2. Eggs per gram (EPG) and proportion (%) of positive animals…, 6) Use only one decimal place,7) Liner 112 – replace rate with value or proportion, 8) Results – use only one decimal place

Answer11: All of them were accepted in the revised MS.

  1. Line 79 – how many pandas were excluded?

Answer12:There are 6 pandas were excluded. Those individuals would move to Care Horse separately.

  1. Table is incomplete for Albendazole – it would be better to exclude this drug

Answer13: The common method of FECR is based on the EPG of Day 15. The fecal egg count reduction% (FECR%) =100(1-Xpost/Xpre) (Xpre: EPG before anthelmin-tic treatment on Day 0; Xpost: EPG post anthelmintic treatment on Day 15). The EPG after treatment with other drugs in the present study were 11 times, but only the EPG on Day 15 was used calculated the FECT to assessing the AR. Although albendazole was incomplete, to assess the AR against B. schroederi is enough only with the EPG on Day 15, we believed that keep it in the MS is better than exclude it. 

  1. Line 113 – not only on Day 15.

Answer14: Although EPG were 11 times after treatment with MBZ, PYR and IVM, but only the EPG on Day 15 was used calculated the FECT to assessing the AR like aforementioned.

  1. 15. Line 115 – which one of the two approaches has been presented in Table 2?

Answer15: The data in the table 2 are mean EPG (n/g) and infection rate (%). Those two approaches are for AR assessing an it was presented in the table 3 (Arithmetic Mean and MCMC Bayesian)

Figure and table

  1. Figure 1 – delete rate; what is the meaning of “g”? Before treatment column is confusing

Answer16: the rate was instead by proportion in the revised MS; The g means gram; “before treatment and after treatment” were replaced by before deworming and after deworming.

  1. Those comments of 1) Table 3. The anthelmintic efficacy assessed by arithmetic mean…, 3) Criteria (instead of criterions); Interpretation…, 3) Line 171 – Current evidence from…

Answer17: all of them were accepted in the revised MS.

  1. Conclusions - they should be more focused on the results.

Answer18: We have revise related content. The paragraph of “Although some measures, such as maintaining good hygiene ….. new animal residents to avoid introducing new sources of B. schroederi.” , has been moved to the section of discussion in the last paragraph in L223-230.

A new conclusion of “MBZ, ABZ and IVM were determined to be efficacious against B. schroederi, while nematode was suspected to be resistant to PYR. The number of eggs per gram in the feces of giant pandas enrolled in the present study were increased near the end of the experiment”was concluded in L232-235.

  1. References – adapt to the style of Animals – not sure whether titles need to be between quotation marks, use lowercase as much as possible, etc.

Answer19: We check and change the references style according the journal require.

  1. Ref. 5 – is there an abstract in English?

Answer20: Yes, this Chinese article has the English abstract.

Round 2

Reviewer 1 Report

I am satisfied with the authors' responses to my comments. Furthermore, the changes made to the manuscript have increased its quality. In my opinion, the manuscript in its present form should be accepted.

Reviewer 3 Report

The authors have addressed all my comments and accepted my suggestions.